# Sufficient dimension reduction for classification using principal optimal transport direction

**Cheng Meng**[1] **Jun Yu**[2] **Jingyi Zhang**[3] **Ping Ma**[4] **Wenxuan Zhong**[*4]

[1] Center for Applied Statistics, Institute of Statistics and Big Data, Renmin University of China
[2] School of Mathematics and Statistics, Beijing Institute of Technology
[3] Center for Statistical Science, Tsinghua University
[4] Department of Statistics, University of Georgia
∗ Corresponding author
chengmeng@ruc.edu.cn, yujunbeta@bit.edu.cn, jingyizhang@tsinghua.edu.cn, pingma@uga.edu,
wenxuan@uga.edu

## Abstract

Sufficient dimension reduction is used pervasively as a supervised dimension reduction approach. Most existing sufficient dimension reduction methods are developed for data with a continuous response and may have an unsatisfactory performance for the categorical response, especially for the binary-response. To address this issue, we propose a novel estimation method of sufficient dimension reduction subspace (SDR subspace) using optimal transport. The proposed method, named principal optimal transport direction (POTD), estimates the basis of the SDR subspace using the principal directions of the optimal transport coupling between the data respecting different response categories. The proposed method also reveals the relationship among three seemingly irrelevant topics, i.e., sufficient dimension reduction, support vector machine, and optimal transport. We study the asymptotic properties of POTD and show that in the cases when the class labels contain no error, POTD estimates the SDR subspace exclusively. Empirical studies show POTD outperforms most of the state-of-the-art linear dimension reduction methods.

## 1 Introduction

Sufficient dimension reduction (SDR) has been one of the most popular linear dimension reduction frameworks in statistics [40, 14, 35]. Given a predictor $X \in \mathbb{R}^p$ and a response $Y \in \mathbb{R}$, sufficient dimension reduction aims to find a projection matrix $\mathbf{B} \in \mathbb{R}^{p \times q}$ ($p \geq q$) such that

$$Y \perp\!\!\!\perp X | \mathbf{B}^T X, \tag{1}$$

where $\perp\!\!\!\perp$ denotes statistical independence. Model (1) indicates that the projected predictor $\mathbf{B}^T X$ preserves all the information about $Y$ contained in $X$. The column space of $\mathbf{B}$, denoted as $\mathcal{S}(\mathbf{B})$ is called a sufficient dimension reduction subspace (SDR subspace). One special property of the sufficient dimension reduction framework is that the model (1) does not assume any specific relationship between $Y$ and $X$. Nowadays, sufficient dimension reduction has played crucial roles in various statistical and machine learning applications, such as classification problems [33], online learning [9], medical research [50], and causal inference [45]. Some popular sufficient dimension reduction techniques include sliced inverse regression (SIR) [40], principal Hessian directions (PHD) [41], sliced average variance estimator (SAVE) [16], directional regression (DR) [38], among others.

Despite its popularity, the success of most existing SDR methods highly depends on the restricted conditions that are imposed on the predictors. In practice, these methods may fail to identify the

true SDR subspace when the conditions are not met. We illustrate such a phenomenon through an example that was originally introduced in [42] as a clustering problem. The example includes two C-shaped trigonometric curves with random Gaussian noise tangle with each other in a two-dimensional subspace embedded in $\mathbb{R}^{10}$. There are two classes, one for each curve:

I) $X_1 = 20\cos\theta + Z_1 + 1$, $X_2 = 20\sin\theta + Z_2$, where $Z_1$, $Z_2$, and $\theta$ are independent generated from $\mathcal{N}(0,1)$, $\mathcal{N}(0,1)$, and $\mathcal{N}(\pi, (0.25\pi)^2)$, respectively; $X_3, \ldots, X_{10}$ are independent generated from $\mathcal{N}(0,1)$.

II) $X_1 = 20\cos\theta + Z_1$, $X_2 = 20\sin\theta + Z_2$, where $Z_1$, $Z_2$, and $\theta$ are independent generated from $\mathcal{N}(0,1)$, $\mathcal{N}(0,1)$, and $\mathcal{N}(0, (0.25\pi)^2)$, respectively; $X_3, \ldots, X_{10}$ are independent generated from $\mathcal{N}(0,1)$.

For each class, we first generate a sample of size 300 and then standardize the sample. It is clear that the SDR subspace of this example is the space spanned by $(e_1, e_2)$, where $e_j$ is a column vector with the $j$-th element being 1 and 0 for the rest. Figure 1(a) shows the first two predictors of the data, where two classes are illustrated by different colors, respectively. Figure 1(b) shows the data projected on the SIR direction. Only one projection direction can be estimated by SIR

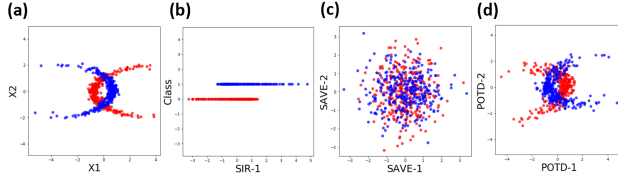

Figure 1: Results for different SDR methods on a 10D binary-response example. The first two predictors of the data are illustrated in panel (a), where different color represent different classes. Panel (b), (c), and (d) show the first one or two directions estimated by SIR, SAVE, and POTD, respectively. POTD is the only one among three that can effectively recover the SDR subspace.

since the response is binary. Figure 1(c) shows the data points projected on the plane spanned by the first two directions estimated by SAVE. Note that both of these two classic SDR techniques fail to provide a decent estimation of the SDR subspace. Such an observation can be attributed to the fact that SIR and SAVE only utilize the first-moment and the second-moment information of the predictors. As a result, they are not able to recover the SDR subspace when the first two moments of the predictors respecting different classes are identical.

**Our contributions.** To overcome the aforementioned limitation of the classic SDR methods, we propose a novel approach for estimating the SDR subspace, primary for the data with a categorical response. The proposed method, named principal optimal transport direction (POTD), forms the basis of the SDR subspace using the principal directions of the optimal transport coupling between the data that are respecting different response categories. POTD does not rely on the moment information of the predictors and is able to provide a more decent estimation of the SDR subspace, as shown in Fig. 1(d). We reveal the close relationship between the proposed method and a well-known SDR approach, named principal support vector machine (PSVM) [36]. This approach first utilizes support vector machine (SVM) to find the optimal hyperplane that separates the data respecting different response categories, then use the normal vectors of this hyperplane to construct the SDR subspace. We demonstrate that the "displacement vectors," which respecting the optimal transport map between the data that are respecting different response categories, are highly consistent with the normal vectors of the aforementioned hyperplane. We propose to use these displacement vectors as surrogates for the desired normal vectors to construct the SDR subspace, thus avoid estimating the optimal hyperplane. Theoretically, we show the proposed method can consistently and exclusively estimate the SDR subspace for the data with a binary-response when the class labels contain no error. To the best of our knowledge, our work is the first approach that can achieve the full SDR subspace instead of a partial SDR subspace obtained by alternative approaches, under mild conditions. We show the advantages of POTD over existing linear dimension reduction methods through extensive simulations. Furthermore, we show the proposed method outperforms several state-of-the-art linear dimension reduction methods in terms of classification accuracy through extensive experiments on various real-world datasets.

## 2 Preliminaries

**Sufficient dimension reduction.** In this paper, we consider the sufficient dimension reduction Model (1) in classification problems, i.e., the response $Y \in \{1, \ldots, k\}$ indicating $k$ classes in the data. Recall in Model (1); sufficient dimension reduction methods aim to seek a set of linear

combinations of $X$, i.e., $\mathbf{B}^T X$, such that the response $Y$ depends on the predictors $X$ only through $\mathbf{B}^T X$. Since the projection matrix $\mathbf{B}$ in Model (1) is not unique, the target of interest in SDR is not on $\mathbf{B}$ itself, but the space spanned by the columns of $\mathbf{B}$, denoted as $\mathcal{S}(\mathbf{B})$. The central subspace, denoted by $\mathcal{S}_{Y|X}$, is the intersection of the spaces spanned by all possible $\mathbf{B}$ that satisfy Model (1) and hence has the minimal dimension among all such $\mathbf{B}$'s (Cook, 1998b). It is known that $\mathcal{S}_{Y|X}$ exists uniquely under mild conditions [14]. We call an SDR method exclusive if it induces an SDR subspace that equals to the central subspace. It is known that some popular SDR methods, e.g., SIR, SAVE, and DR, are exclusive under certain conditions [35].

There is a vast literature on SDR methods [46, 82, 47, 79], the majority of which are developed for data with continuous response. Although some of them work fairly well for categorical response data, the performance of some classic SDR methods degrades significantly for binary-response data. For example, SIR can identify at most one direction in binary classification since there are only two slices available. As a result, SIR is unable to recover the full SDR subspace when one direction is insufficient to separate two classes, say the example in Fig.1. For another example, SAVE is known for its inefficient estimation when the response is binary [38], and we refer to [15] for more discussion. There exist other SDR methods in the literature that are designed for categorical response data [60, 61]. The performance of these methods, however, depends on relatively strong assumptions that are imposed on the predictors, and they may not recover the full SDR subspace in some applications. There also exist some nonlinear dimension reduction methods to tackle the classification problems [36, 66]. These nonlinear methods are beyond the scope of this paper.

**Optimal transport methods.** To overcome the aforementioned limitations of the existing SDR methods, we develop a novel SDR approach for data with a categorical response, utilizing optimal transport methods. Optimal transport has been widely studied in mathematics, probability, and economics [25, 63, 59]. Recently, as a powerful tool to transform one probability measure to another, optimal transport methods find extensive applications in machine learning [3, 18, 55, 4, 10, 27, 48, 53, 77, 78, 68, 67, 26, 58], statistics [21, 11, 51, 72, 49, 54, 20, 28, 17], computer vision [25, 56, 63, 55, 22, 69], and so on.

Let $\mathscr{P}(\mathbb{R}^p)$ be the set of Borel probability measures in $\mathbb{R}^p$, and let

$$\mathscr{P}_2(\mathbb{R}^p) = \left\{ \mu \in \mathscr{P}(\mathbb{R}^p) \,\middle|\, \int \|\boldsymbol{x}\|^2 d\mu(\boldsymbol{x}) < \infty \right\}.$$

Consider two probability measures $\mu, \nu \in \mathscr{P}_2(\mathbb{R}^p)$. For any measurable set $\Omega \subset \mathbb{R}^p$, we define $\phi_{\#}(\mu)(\Omega) = \mu(\phi^{-1}(\Omega))$. Let $\Phi$ be the set of all the so-called measure-preserving map $\phi: \mathbb{R}^p \to \mathbb{R}^p$, such that $\phi_{\#}(\mu) = \nu$ and $\phi_{\#}^{-1}(\nu) = \mu$. Let $\| \cdot \|$ be the Euclidean norm. Among all the maps in $\Phi$, the optimal one respecting to $L_2$ transport cost is defined as

$$\phi^* = \inf_{\phi \in \Phi} \int_{\mathbb{R}^p} \|\boldsymbol{a} - \phi(\boldsymbol{a})\|^2 \mathrm{d}\mu(\boldsymbol{a}). \tag{2}$$

Formulation (2) is usually called the Monge formulation and its solution $\phi^*$ is usually called the optimal transport map, or Monge map. The vector $\boldsymbol{a} - \phi^*(\boldsymbol{a})$ is called the displacement vector of the optimal transport map $\phi^*$ on $\boldsymbol{a}$. One limitation for the Monge formulation (2) is that, it may be infeasible in some extreme cases, say, when $\mu$ is a Dirac measure but $\nu$ is not. To overcome such an limitation, Kantorovich considered the following set of "couplings" [64],

$$\Pi(\mu, \nu) = \{ \pi \in \mathscr{P}(\mathbb{R}^p \times \mathbb{R}^p) \quad s.t. \quad \forall \quad \text{Borel set} \quad A, B \subset \mathbb{R}^p,$$
$$\pi(A \times \mathbb{R}^p) = \mu(A), \quad \pi(\mathbb{R}^p \times B) = \nu(B) \}.$$

Kantorovich then formulated the optimal transport problem as finding the optimal coupling, i.e., the joint probability measure $\pi$ from $\Pi(\mu, \nu)$, that minimizes the expected cost,

$$\pi^* = \inf_{\pi \in \Pi(\mu,\nu)} \int \|x - y\|^2 \mathrm{d}\pi(x, y). \tag{3}$$

Formulation (3) is usually called the Kantorovich formulation and its solution $\pi^*$ is called the optimal transport plan or optimal coupling. Consider the cases when both $\mu$ and $\nu$ are continuous probability measures that vanish outside a compact set, and both measures have continuous densities with respect to the Lebesgue measure. Under the above setups, the celebrated Brenier theorem [7] guarantees the existence of the Monge map, and it is shown that the solution of the Monge formulation (2) is

equivalent to the solution of the Kantorovich formulation (3) [32, 31]. For mathematical simplicity, we assume the Monge map exists when studying the theoretical properties of the proposed method. However, the proposed algorithm is developed upon the optimal coupling, and it does not require the existence of the Monge map. Furthermore, the numerical results in Section 5 indicate the proposed method empirically works well when the Monge map does not exist.

## 3  Motivation and Methodology

To motivate the development of the proposed method, we first re-examine a popular SDR approach, named principal support vector machine (PSVM). Such an approach is first proposed in [36], and is further developed in [5, 80, 60, 61]. Recall that SVM is a classification method that draws an optimal hyperplane to separate the data respecting different categories. Consider the data with the binary-response. [36] first observed that the normal vectors of the hyperplane that learned by SVM, are as much as possible aligned with the directions in which the regression surface varies, i.e., the directions that form the SDR subspace. Consequently, these normal vectors can be naturally utilized to construct the SDR subspace. In particular, the authors in [36] proposed to combine these normal vectors by principal component analysis to form the basis of the SDR subspace.

We provide an example to illustrate the idea of PSVM. Fig. 2(a) shows a synthetic 3D sample with a binary-response, and the subsample respecting different response categories are marked in blue and red, respectively. For these two subsamples, their marginal distributions of $e_1$ have the same mean and different variances, their marginal distributions of $e_2$ have the same variance and different means, and their marginal distributions of $e_3$ are the same. Consequently, the SDR subspace of this example is spanned by $(e_1, e_2)$.

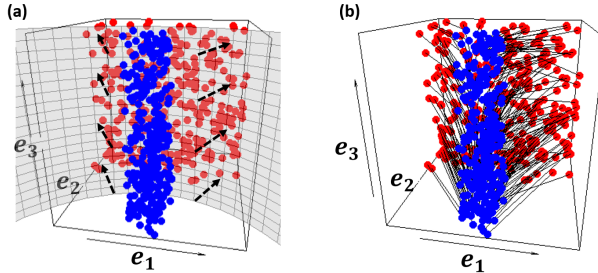

Figure 2: Illustration for PSVM and POTD on a 3D synthetic binary-response dataset. The data respecting difference response categories are marked by blue and red, respectively. Panel (a) shows the hyperplane learned by SVM, and the normal vectors of this hyperplane are labeled as dashed arrows. Panel (b) shows all the displacement vectors of the optimal transport map between two classes. We observe that the displacement vectors in (b) show highly consistent patterns as the normal vectors in (a).

The PSVM approach first uses SVM to calculate the optimal hyperplane that separates the two classes. The hyperplane is illustrated by the grey curved surface in Fig. 2(a). Using the principal component analysis, the normal vectors of this hyperplane, shown as the dashed arrows, are then used to form the basis of the SDR subspace. The PSVM approach is superior to the classic SIR and SAVE method in this example. Note that SIR and SAVE only utilizes the first-moment and the second-moment information of the predictors, respectively. Neither SIR and SAVE could successfully recover the full SDR subspace in this example, since the former can only identify the subspace spanned by $e_2$ and the latter can only identify the subspace spanned by $e_1$. Despite the advantages, one limitation for the PSVM approach is that the explicit form of the desired normal vectors are only available for linear SVM but are not available for kernel SVM. Although several nonlinear dimension reduction methods are developed, still lacking are the linear dimension reduction methods for the cases when the data respecting different response categories are not linearly separable, e.g., the example in Fig.1(a).

To address the aforementioned issue, we propose to utilize the optimal transport technique to construct surrogates for the desired normal vectors. The key intuition is that, the displacement vectors of the optimal transport map that maps one subsample to the other subsample are as much as possible orthogonal to the hyperplane that can best separate these two subsamples. As a result, these displacement vectors are highly consistent with the normal vectors of the hyperplane estimated by SVM. For illustration, we plot the displacement vectors respecting the synthetic dataset in Fig. 2(b), and it is clear that these vectors show highly consistent patterns as the normal vectors in Fig. 2(a). Consequently, using optimal transport, we can obtain the desired vectors directly and explicitly, without calculating the optimal hyperplane. Analogous to the PSVM approach, we propose to combine the displacement vectors by principal component analysis to form the basis of the SDR subspace. To extend the proposed method for binary-response cases to multi-class response cases,

two popular strategies could be used, namely the one-vs-one strategy and the one-vs-rest strategy. We opt to adopt the former in this paper, and we find using the latter yields similar results.

We now provide more details of the proposed method. Let $k \geq 2$ be the number of response categories, $\mathbf{X} \in \mathbb{R}^{n \times p}$ be the pooled sample matrix, and we assume that $\mathbf{X}^T\mathbf{X} = \mathbf{I}_p$, without lose of generality. Let $n_i$ be the number of samples respecting to the $i$-th class, $\mathbf{X}_{(i)} \in \mathbb{R}^{n_i \times p}$ be the subsample matrix respecting to the $i$-th class, and $\boldsymbol{a}_i = (a_{i1}, \ldots, a_{in_i})^\intercal$ be the given weight vector corresponding to $\mathbf{X}_{(i)}$. Without loss of generality, we assume the $L_1$ norm of $\boldsymbol{a}_i$ equals one, $i = 1, \ldots, k$. Such an assumption ensures that the optimal transport problems, between the data respecting difference response categories, are valid. In the cases when $\{\boldsymbol{a}_i\}_{i=1}^k$ do not have the same value, one can simply replace $\{\boldsymbol{a}_i\}_{i=1}^k$ by $\{\boldsymbol{a}_i/\|\boldsymbol{a}_i\|_1\}_{i=1}^k$ before implementing the proposed method.

For any $i, j \in \{1, \ldots, k\}$, we calculate the empirical optimal transport coupling matrix $\mathbf{G}_{ij} \in \mathbb{R}^{n_i \times n_j}$ between the weighted samples $(\mathbf{X}_{(i)}, \boldsymbol{a}_i)$ and $(\mathbf{X}_{(j)}, \boldsymbol{a}_j)$. The matrix $\mathbf{G}_{ij}$ is then used to construct the "displacement matrix" $\boldsymbol{\Delta}_{ij} \in \mathbb{R}^{n_i \times p}$,

$$\boldsymbol{\Delta}_{ij} = \operatorname{diag}(\boldsymbol{a}_i)\mathbf{X}_{(i)} - \mathbf{G}_{ij}\mathbf{X}_{(j)},$$

where $\operatorname{diag}(\boldsymbol{a}_i) \in \mathbb{R}^{n_i \times n_i}$ is the diagonal matrix such that its diagonal vector equals $\boldsymbol{a}_i$. In the literature, $\mathbf{G}_{ij}\mathbf{X}_{(j)}$ is usually termed the barycentric projection of $\mathbf{X}_{(j)}$. In particular, let $\widehat{\phi}^*$ be the empirical optimal transport map between $(\mathbf{X}_{(i)}, \boldsymbol{a}_i)$ and $(\mathbf{X}_{(j)}, \boldsymbol{a}_j)$, if it exists. It is thus easy to check that, for $l = 1, \ldots, n_i$, the $l$-th row of $\boldsymbol{\Delta}_{ij}$ equals $a_{il}(\mathbf{X}_{(i)l} - \widehat{\phi}^*(\mathbf{X}_{(i)l}))^\intercal$, which is the displacement vector of the $l$-th observation in $\mathbf{X}_{(i)}$ weighted by $a_{il}$.

Next, to apply the one-vs-one strategy, we let $\boldsymbol{\Lambda}_{(i)} \in \mathbb{R}^{n_i(k-1) \times d}$ be the matrix that vertically stacks all the $\boldsymbol{\Delta}_{ij}$s, $j = 1, \ldots, i-1, i+1, \ldots, k$. Furthermore, we let $\boldsymbol{\Lambda} \in \mathbb{R}^{n(k-1) \times d}$ be the matrix that vertically stacks all the $\boldsymbol{\Lambda}_{(i)}$s, $i = 1, \ldots, k$. The final SDR subspace then can be constructed using the leading right singular vectors of $\boldsymbol{\Lambda}$. Algorithm 1 summarizes the proposed method.

---

**Algorithm 1** Principal Optimal Transport Direction (POTD)

---

**Input:** $\mathbf{X} \in \mathbb{R}^{n \times d}, Y \in \{1, \ldots, k\}, \boldsymbol{a} \in \mathbb{R}^n$, the structure dimension $r$
**for** $i$ in $1:k$ **do**
    **for** $j$ in $\{1, \ldots, i-1, i+1, \ldots, k\}$ **do**
        $\mathbf{G}_{ij} \leftarrow \mathrm{OT}[(\mathbf{X}_{(i)}, \boldsymbol{a}_i), (\mathbf{X}_{(j)}, \boldsymbol{a}_j), \mathrm{cost} = \|\cdot\|^2]$
        $\boldsymbol{\Delta}_{ij} = \operatorname{diag}(\boldsymbol{a}_i)\mathbf{X}_{(i)} - \mathbf{G}_{ij}\mathbf{X}_{(j)}$
    **end for**
    $\boldsymbol{\Lambda}_{(i)} = \begin{pmatrix} \boldsymbol{\Delta}_{i1} \\ \ldots \\ \boldsymbol{\Delta}_{ij} \end{pmatrix}$
**end for**
$\boldsymbol{\Lambda} = \begin{pmatrix} \boldsymbol{\Lambda}_{(1)} \\ \ldots \\ \boldsymbol{\Lambda}_{(k)} \end{pmatrix}$
**Output:** $\mathbf{v}_1, \ldots, \mathbf{v}_r$, i.e., the first $r$ right singular vectors of $\boldsymbol{\Lambda}$

---

**Computational cost.** In practice, the coupling matrix $\mathbf{G}_{ij}$ in Algorithm 1 can be calculated using the Sinkhorn algorithm [19]. In the cases when all $k$ classes have roughly similar sample sizes, the computational cost for calculating $\mathbf{G}_{ij}$ is at the order of $O((n/k)^2 \log(n/k)p)$. The total number of coupling matrices that need to be calculated is $k(k-1)$, and thus the overall computational cost for optimal transport is of the order $O(n^2 \log(n)p)$. Furthermore the computational cost for calculating the SVD for the matrix $\boldsymbol{\Lambda}$ is at the order of $O(knp^2)$. Thus, the overall computational cost for Algorithm 1 is $O(n^2 \log(n)p + knp^2)$. Besides, one may use other optimal transport algorithms, like Greenkhorn [2] and Nys-Sink [1], for potentially faster calculations.

**Estimation of structure dimension.** In Algorithm 1, we assume the structure dimensional $r$ is known. This information, however, may not be available in practice. In the literature of sufficient dimension reduction, there exist several methods to determine $r$. For example, a chi-squared test was developed in the SIR method [40]. However, the extension of the test beyond SIR is still lacking. The BIC-type approach is another widely-used procedure [81, 82]. Neither of these approaches is applicable to our algorithm since they are developed based on the asymptotic normality, which is

rarely the case in our setting. Based on our empirical studies, we suggest using the cumulative ratio of the singular values to choose the structure dimension [30]. Such a procedure is commonly used by classical dimension reduction methods such as principal component analysis.

# 4   Theoretical results

For ease of exposition, throughout this section, we only consider the sufficient dimension reduction model (1) with binary-response cases, i.e., $Y \in \{0, 1\}$. We also assume the optimal transport map exists. We propose a predictor of $Y$, called $Y^*$, and study some properties of $Y^*$ from the perspective of the optimal transport theory.

Now we reformulate the binary-response sufficient dimension reduction problem as a transport problem. Without loss of generality, one class of sample is called the source sample, and the other class of sample is the target sample. Given the pooled sample consisting of both the source sample and the target sample, we randomly take a sample point, say $X$, and its associated label (response) $Y$, which equals zero if the sample point $X$ is from the source sample and one otherwise. Here, $Y$ follows a Bernoulli distribution, where $Y$ is equal to one with the probability $P(X)$. Given $P(\cdot)$, we employ the predictor $Y^* = I(P(X) \geq 0.5)$, where $I(\cdot)$ is the indicator function, to predict label $Y$ of the corresponding sample point $X$. Obviously, for error-degenerated cases, i.e., when all class labels have been correctly predicted, we have $Y^* = Y$.

Analogous to the central subspace $\mathcal{S}_{Y|X}$, we now define the subspace $\mathcal{S}_{Y^*|X}$. Suppose there exists some $\mathbf{B} \in \mathbb{R}^{p \times r}$ such that
$$Y^* \perp\!\!\!\perp X | \mathbf{B}^T X. \tag{4}$$
The subspace $\mathcal{S}_{Y^*|X}$ is defined as the intersection of the spaces spanned by all possible $\mathbf{B}$ that satisfy equation (4). Note that one has $\mathcal{S}_{Y^*|X} \subseteq \mathcal{S}_{Y|X}$, since $Y^*$ can be represented as a function of $\mathbb{E}(Y|X)$, which equals $P(X)$. Consider the case that the binary-response $Y$ is a function $Y(Y^*, \epsilon)$ of $Y^*$ conditioning on $X = x$ and a random variable $\epsilon$, which follows a Bernoulli distribution and is independent of the predictor $X$, indicates whether the response has been correctly observed as the true class label. We thus have $\mathcal{S}_{Y|X} \subseteq \mathcal{S}_{Y^*|X}$. Consequently, $Y$ and $Y^*$ contain exactly the same information about $X$ in this situation. This is to say, we have $\mathcal{S}_{Y^*|X} = \mathcal{S}_{Y|X}$ under the condition
$$Y \perp\!\!\!\perp X | Y^*. \tag{5}$$
Specifically, for the binary outcome data, when one explanatory variable or a combination of explanatory variables can perfectly predicts all the labels, condition (5) naturally holds. This is known as the "separation" property in statistics, and we refer to [34, 29, 57] for more detailed discussion and general view of the concept of "separation". Consequently, we have $\mathcal{S}_{Y^*|X} = \mathcal{S}_{Y|X}$ as long as the data enjoys the "separation" property.

Let $\mu$ and $\nu$ be the probability measures of $X|Y^* = 1$ and $X|Y^* = 0$, respectively. Let $\phi^*$ be the optimal transport map from the set $\Phi = \{\phi : \mathbb{R}^p \to \mathbb{R}^p | \phi_\#(\mu) = \nu; \phi_\#^{-1}(\nu) = \mu\}$, respecting the $L_2$ norm. We consider the so-called "second-order displacement matrix" of $\phi^*$, which is defined as
$$\mathbf{\Sigma} = \int \Big( (I - \phi^*)(X) \Big) \Big( (I - \phi^*)(X) \Big)^T d\mu(X). \tag{6}$$
Let $\lambda_1 \geq \ldots \geq \lambda_p$ be the eigenvalues of $\mathbf{\Sigma}$. Given an integer $c \leq p$, let $\mathbf{V}_c$ be the matrix whose columns are the first $c$ eigenvectors of matrix $\mathbf{\Sigma}$, i.e., $\mathbf{V}_c = (v_1, \ldots, v_c) \in \mathbb{R}^{p \times c}$ and its orthonormal columns satisfy $\mathbf{\Sigma} v_j = \lambda_j v_j$ for $j = 1, \ldots, c$. To avoid trivial cases, we only consider the scenarios that $p > 4$ in this section. We now present some essential regularity conditions for our main results.

**(H.1)** The subspace $\mathcal{S}_{Y^*|X}$ exists and is unique.

**(H.2)** Let $r$ be the structure dimension, i.e., the dimension of $\mathcal{S}_{Y^*|X}$. Suppose $\lambda_r - \lambda_{r+1} \geq \delta$ for some $\delta > 0$, and $\lambda_{p+1} = -\infty$ for simplification.

**(H.3)** Suppose $\{(Y_i, X_i)\}_{i=1}^n$ are independent and identically distributed. The probability distributions of both $X|Y^* = 1$ and $X|Y^* = 0$ have positive densities in the interior of their convex supports and have finite moments of order $4 + \delta$ for some $\delta > 0$.

**(H.4)** Let $N(\mu, \epsilon, \tau)$ be the minimal number of $\epsilon$-balls whose union has $\mu$ measure at least $1 - \tau$. We assume $N(\mu, \epsilon, \epsilon^{2p/(p-4)}) \leq \epsilon^{-p}$ and $N(\nu, \epsilon, \epsilon^{2p/(p-4)}) \leq \epsilon^{-p}$.

**(H.5)** Let $n_0$ and $n_1$ be the number of observations such that $Y^* = 0$ and $Y^* = 1$, respectively. We assume $n_0/(n_0 + n_1) \to C$, for some constant $C \in (0, 1)$, as $n = n_0 + n_1 \to \infty$.

Condition (H.1) naturally holds when classification probability $P(X)$ can be represented as a function of $\mathbf{B}^T X$, for some $\mathbf{B} \in \mathbb{R}^{p \times r}$, up to some unknown latent factors which are independent of $X$. This is quite common under the logistic regression setting. The so-called eigen-gap condition (H.2), ensuring that the signal associated with the largest $r$ eigenvalues is separable from the noise associated with the rest eigenvalues, is a quite common assumption in the statistical learning literature, see [71]. Conditions (H.3)–(H.5) are the convergence conditions of the estimated optimal transport map and are widely used in the optimal transport theory, see [21, 51]. We now present our main theorem, the proof of which is relegated to the Supplementary Material.

**Theorem 1.** *Let $\phi^*$ be the optimal transport map under the assumptions (H.1) – (H.2), we have*

$$\mathcal{S}(\mathbf{V}_r) = \mathcal{S}_{Y^*|X} \subseteq \mathcal{S}_{Y|X},$$

*where $\mathcal{S}(\mathbf{V}_r)$ is the column space of a matrix $\mathbf{V}_r$.*

Theorem 1 indicates that one can recover the subspace $\mathcal{S}_{Y^*|X}$ via the column space of $\mathbf{V}_r$, i.e, the space spanned by the first $r$ eigenvectors of $\boldsymbol{\Sigma}$. Recall that we have $\mathcal{S}_{Y^*|X} = \mathcal{S}_{Y|X}$ as long as the class labels contain no error, or the data enjoys the "separation" property. In those cases, Theorem 1 provides a theoretical guarantee to recover the SDR subspace exclusively.

In practice, optimal transport map $\phi^*$ and matrix $\boldsymbol{\Sigma}$ need to be estimated from the data. Let $\widehat{\phi}^*$ be the estimated optimal transport map, $\widehat{\boldsymbol{\Sigma}} = n_1^{-1}(\mathbf{X} - \widehat{\phi}^*(\mathbf{X}))^T(\mathbf{X} - \widehat{\phi}^*(\mathbf{X}))$, be the empirical second-order displacement matrix with eigenvalues $\widehat{\lambda}_1 \geq \ldots \geq \widehat{\lambda}_p$, and $\widehat{\mathbf{V}}_r = (\widehat{\boldsymbol{v}}_1, \ldots, \widehat{\boldsymbol{v}}_r) \in \mathbb{R}^{p \times r}$ be the matrix whose columns are the first $r$ eigenvectors of $\widehat{\boldsymbol{\Sigma}}$.

**Theorem 2.** *Assume assumptions (H.1)–(H.5) hold. We have*

$$\| \sin(\mathbf{V}_r, \widehat{\mathbf{V}}_r) \|_F := \| \boldsymbol{P}_{\mathbf{V}_r} \boldsymbol{P}_{\widehat{\mathbf{V}}_r^\perp} \|_F = O_p(n^{-1/p}),$$

*where $\| \cdot \|_F$ denotes the Frobenius norm, $\boldsymbol{P}_{\mathbf{V}_r}$ is the projection matrix on the column space of $\mathbf{V}_r$, and $\boldsymbol{P}_{\widehat{\mathbf{V}}_r^\perp}$ is the projection matrix on the orthogonal complement of column space of $\widehat{\mathbf{V}}_r$.*

Theorem 2 states that column space of $\widehat{\mathbf{V}}_r$, the space spanned by eigenvectors of $\widehat{\boldsymbol{\Sigma}}$, converges to the column space of $\mathbf{V}_r$, the central dimension reduction subspace $S_{Y^*|X}$.

Based on the discussion above, the space $\mathcal{S}_{Y^*|X}$ can be recovered by the optimal transport map $\phi^*$ and its corresponding displacement matrix $\boldsymbol{\Sigma}$. Note that when $Y^* = Y$, the space $\mathcal{S}_{Y^*|X}$ is equivalent to the space $\mathcal{S}_{Y|X}$. Recall that we have $Y^* = Y$ in the error-degenerated cases, which are quite common in practice. For example, in image classification, one may assume all images are correctly labeled. For simplicity, we assume $Y^* = Y$ in the next section.

## 5   Numerical experiments

**Simulation studies.** We evaluate the finite sample performance of the proposed POTD method on synthetic data. For comparison, we consider fifteen popular supervised linear dimension reduction methods: AMMC[44], ANMM[65], DAGDNE[23], DNE[74], ELDE[24], LDE[13], LDP[75], LPFDA[76], LSDA[8], MMC[39], MODP[73], MSD[62], ODP[37], SAVE[16], PHD[41], where the first thirteen are implemented in R package `Rdimtools` and the last two are implemented in `dr`. All parameters are set as default. We did not consider SIR [40] here since the dimension of the true SDR subspace in our studies is larger than one, while SIR can estimate at most one direction for binary classification problems.

Throughout the simulation, we set $n = 400$ and $p = 10, 20, 30$. We generate the binary-response data from the following four models:

I:   $Y = \text{sign}\{\sin(X_1)/X_2^2 + 0.2\epsilon\}$;
II:  $Y = \text{sign}\{(X_1 + 0.5)(X_2 - 0.5)^2 + 0.2\epsilon\}$;
III: $Y = \text{sign}\{\log(X_1^2)(X_2^2 + X_3^2/2 + X_4^2/4) + 0.2\epsilon\}$;
IV:  $Y = \text{sign}\{\sin(X_1)/(X_2 X_3 X_4) + 0.2\epsilon\}$;

Table 1: Averaged space distances over 100 independent (lower the better), with standard deviations presented in parentheses. The best result for each scenario is marked in bold.

| Model-$p$ | AMMC | ANMM | DAGDNE | DNE | ELDE | LDE | LDP | LPFDA |
|---|---|---|---|---|---|---|---|---|
| I-10 | 1.73(0.23) | 0.97(0.12) | 0.98(0.08) | **0.37**(0.13) | 1.05(0.13) | 1.82(0.22) | 1.70(0.12) | 1.65(0.21) |
| II-10 | 1.73(0.23) | 0.95(0.15) | 0.91(0.12) | 0.73(0.27) | 0.96(0.19) | 1.83(0.13) | 1.77(0.13) | 1.47(0.23) |
| III-10 | 2.26(0.22) | 2.42(0.47) | 1.99(0.21) | 2.15(0.22) | 1.99(0.31) | 2.78(0.31) | 2.61(0.26) | 2.34(0.31) |
| IV-10 | 2.29(0.24) | 2.46(0.31) | 2.19(0.24) | 1.70(0.47) | 2.23(0.35) | 2.79(0.41) | 2.46(0.22) | 2.33(0.31) |
| I-20 | 1.86(0.08) | 1.10(0.10) | 1.10(0.05) | 1.01(0.14) | 1.78(0.27) | 1.84(0.12) | 1.89(0.06) | 1.20(0.08) |
| II-20 | 1.90(0.09) | 1.05(0.05) | 1.04(0.06) | 0.99(0.09) | 1.50(0.34) | 1.84(0.07) | 1.89(0.07) | 1.06(0.08) |
| III-20 | 3.19(0.21) | 3.14(0.26) | 2.65(0.26) | 2.62(0.12) | 2.96(0.21) | 3.30(0.17) | 3.33(0.24) | 3.25(0.18) |
| IV-20 | 3.21(0.22) | 3.13(0.24) | 3.16(0.20) | 3.07(0.21) | 3.10(0.31) | 3.29(0.23) | 3.28(0.23) | 3.25(0.21) |
| I-30 | 1.90(0.04) | 1.18(0.09) | 1.14(0.13) | 1.16(0.13) | 1.66(0.26) | 1.93(0.05) | 1.93(0.04) | 1.15(0.07) |
| II-30 | 1.92(0.05) | 1.10(0.04) | 1.11(0.03) | 1.10(0.03) | 1.52(0.35) | 1.93(0.04) | 1.92(0.03) | 1.05(0.04) |
| III-30 | 3.48(0.15) | 3.34(0.19) | 2.86(0.10) | 2.83(0.08) | 3.44(0.17) | 3.47(0.17) | 3.47(0.08) | 3.49(0.15) |
| IV-30 | 3.50(0.16) | 3.49(0.17) | 3.49(0.13) | 3.37(0.13) | 3.52(0.13) | 3.57(0.13) | 3.52(0.15) | 3.47(0.12) |

| Model-$p$ | LSDA | MMC | MODP | MSD | ODP | SAVE | PHD | POTD |
|---|---|---|---|---|---|---|---|---|
| I-10 | 1.02(0.04) | 1.72(0.22) | 1.55(0.22) | 1.72(0.21) | 1.55(0.14) | 1.06(0.15) | 1.68(0.21) | 0.66(0.15) |
| II-10 | 0.90(0.17) | 1.70(0.23) | 1.55(0.21) | 1.72(0.21) | 1.55(0.17) | 0.88(0.32) | 1.36(0.21) | **0.69**(0.19) |
| III-10 | 2.02(0.29) | 2.26(0.28) | 2.26(0.29) | 2.26(0.29) | 2.27(0.24) | 2.01(0.24) | 2.01(0.29) | **1.61**(0.18) |
| IV-10 | 2.30(0.29) | 2.28(0.29) | 2.27(0.29) | 2.28(0.29) | 2.27(0.30) | 2.39(0.30) | 2.39(0.29) | **1.39**(0.26) |
| I-20 | 1.08(0.06) | 1.86(0.09) | 1.76(0.09) | 1.87(0.06) | 1.76(0.24) | 1.55(0.07) | 1.85(0.06) | **0.85**(0.07) |
| II-20 | 1.03(0.03) | 1.89(0.09) | 1.76(0.09) | 1.90(0.06) | 1.76(0.11) | 1.00(0.10) | 1.79(0.06) | **0.84**(0.07) |
| III-20 | 2.61(0.12) | 3.25(0.23) | 3.22(0.23) | 3.22(0.24) | 3.22(0.16) | 2.49(0.16) | 2.48(0.24) | **2.12**(0.19) |
| IV-20 | 3.15(0.19) | 3.23(0.24) | 3.22(0.24) | 3.23(0.24) | 3.22(0.18) | 3.18(0.18) | 3.18(0.24) | **2.36**(0.26) |
| I-30 | 1.14(0.08) | 1.90(0.05) | 1.85(0.05) | 1.90(0.05) | 1.85(0.12) | 1.82(0.06) | 1.92(0.05) | **1.00**(0.06) |
| II-30 | 1.07(0.03) | 1.92(0.05) | 1.85(0.06) | 1.92(0.05) | 1.85(0.10) | 1.13(0.08) | 1.89(0.05) | **0.91**(0.05) |
| III-30 | 2.82(0.11) | 3.49(0.15) | 3.48(0.15) | 3.48(0.15) | 3.48(0.10) | 2.74(0.10) | 2.74(0.15) | **2.52**(0.13) |
| IV-30 | 3.51(0.14) | 3.50(0.09) | 3.50(0.09) | 3.50(0.09) | 3.50(0.18) | 3.47(0.17) | 3.48(0.09) | **2.80**(0.13) |

where $X = (X_1, \ldots, X_4)$ follows the multivariate uniform distribution UNIF$[-2, 2]^p$ and $\epsilon$ follows standard normal distribution. Recall that $e_j$ is a column vector with the $j$-th element being 1 and 0 for the rest. The true SDR subspace $\mathcal{S}(\mathbf{B}_0)$ is spanned by $(e_1, e_2)$ for model I and model II, and is spanned by $(e_1, e_2, e_3, e_4)$ for model III and model IV. The true structure dimension is assumed to be known. We use the following metric, developed in [70], to measure the distance between the estimated SDR subspace $\mathcal{S}(\widehat{\mathbf{B}})$ and the true SDR subspace $\mathcal{S}(\mathbf{B}_0)$,

$$m(\mathcal{S}(\widehat{\mathbf{B}}), \mathcal{S}(\mathbf{B}_0)) = ||(\mathbf{I}_p - \widehat{\mathbf{B}}\widehat{\mathbf{B}}^T)\mathbf{B}_0||_F.$$

That is to say, a smaller value of the distance associates with a more accurate estimation. Empirically, we find other metrics, like the sine metric considered in Theorem 2, also yield similar performance. The average space distances (the standard deviations are in parentheses) over 100 independent replications are shown in Table 1. The smallest distance (the best result) in each setting is highlighted in bold. We observe that the proposed POTD method provides the best results in all settings except the first set, where the POTD provides the second-best result.

**MNIST data visualization.** We now evaluate the performance of the POTD method as a data visualization tool. We apply it to the MNIST dataset, which contains 60,000 training images and 10,000 testing images of handwritten digits. We shrink each $28 \times 28$ image to $14 \times 14$ using max-pooling and then stack

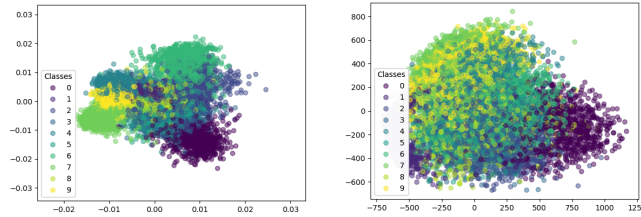

Figure 3: Visualization of MNIST using POTD (left) and PCA (right).

it into a 196-dimensional vector. Figure 3 displays the 2D embeddings of the testing images in the SDR subspace learned using the proposed POTD method (the left panel) and in the first two principal components using PCA from the training images. The colors encode different digit categories (which are not used for training but for visualization). As expected, as a supervised dimension reduction method, POTD yields reasonably better clusters of the data than PCA.

**Classification on real-world datasets.** We now compare POTD with its competitors in terms of the classification accuracy on various real-world datasets. We consider seven multi-class real-world

datasets[1]: *Breast Cancer Wisconsin* (WDBC), *Letter Recognition* (LETTER), *Pop failures* (POP), *QSAR biodegradation* (BIODEG), *Connectionist Bench Sonar* (SONAR), and *Optical Recognition of Handwritten Digits* (OPTDIGITS). From the dataset of *Letter Recognition*, which is a multi-class dataset with 26 class labels, we take a sub-dataset consisting of letters { "D", "O", "Q", "C" } (LETTER1) and a second sub-dataset consisting of letters { "M", "W", "U", "V" } (LETTER2). Each of the two sub-datasets has four letters that are relatively difficult to distinguish.

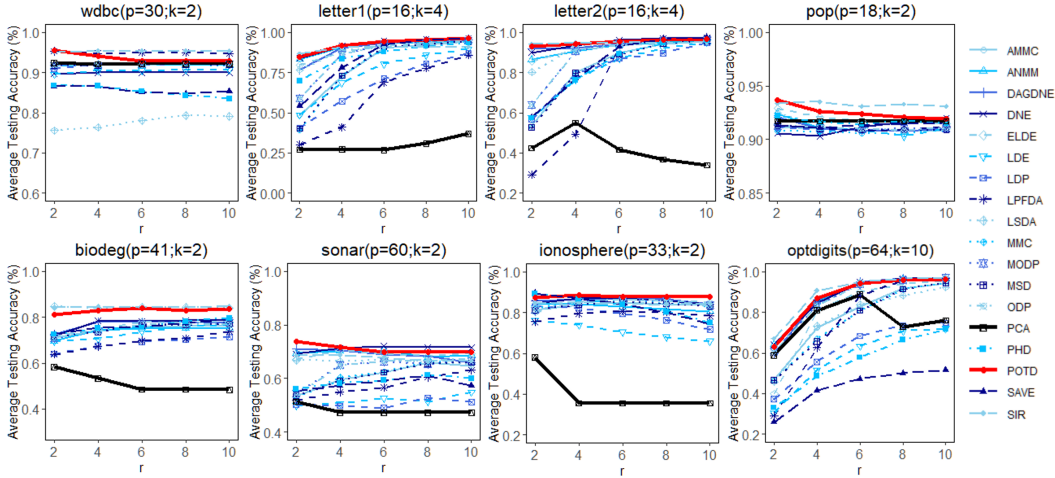

Figure 4: The average testing accuracy on different datasets.

For each dataset, we replicate the experiment one hundred times. In each replication, each dataset is randomly divided into the training set and the testing set of equal sizes. We compare POTD with all the aforementioned fifteen competitors, where all the parameters are set as default. In addition, we also consider the standard SIR method, where the structure dimension for SIR is set to be $\min(r, k-1)$, where $k$ denotes the number of class labels in the training set. This is because the structure dimension estimated by SIR is always smaller than $k$. Moreover, we also use PCA as a baseline method. For all other methods, we consider five different choices of structure dimension $r$, i.e., $r = \{2, 4, 6, 8, 10\}$. For each $r$, the training set is first projected to a $r$-dimensional subspace. We then apply $K$-nearest neighbor classifier to the projected training set, and $K$ is set to be 10. We evaluate the performance of all methods via the $K$-nearest neighbor classifier's average testing accuracy, i.e., $(TP + FN)/n_{test}$, where $TP$ and $FN$ denote true positive and false negative, respectively, and $n_{test}$ is the sample size of the testing set. Empirically, we find the results remain stable for a wide range of $K$.

Figure 4 shows the average testing accuracy versus different sizes of the structure dimension. The dimension $p$ and the number of class label $k$ respecting each dataset are listed in the subtitles. We first observe that occasionally, PCA yields better performance than some of the supervised dimension reduction methods. For the five datasets where the number of classes $k = 2$, we observe that the method of SIR usually gives the best result. As discussed before, SIR can only estimate one sufficient dimension reduction direction in these datasets, no matter how large the structure dimension $r$ is. As it may indicate in these five datasets, the extra directions estimated by other dimension reduction methods may not be necessary to be beneficial for the downstream classifier's classification accuracy. Nevertheless, the extra directions may have potential benefits on data visualization or other downstream quantitative analysis. Finally, we observe that the proposed POTD method performs consistently better than PCA, and it outperforms most of its competitors in all cases. These results demonstrate that POTD is very effective in estimating the SDR subspace for the data with a categorical response. Note that the classification problem considered here is a favorable case for dimension reduction; thus, it warrants the asymptotic convergence of all dimension reduction methods. The results in Fig. 4 indicate that even for such a simple setup, our method can work as well as its competitors.

## Broader Impact

In this paper, we study the problem of sufficient dimension reduction for classification. We propose a novel method to estimate the SDR subspace using the principal directions of the empirical optimal transport plans. The proposed POTD method can consistently and exclusively estimate the SDR subspace for the data with a binary-response when the class labels contain no error, or the data enjoys the "separation" property. The proposed method could be naturally extended to the data with continuous response. In such cases, we can first form several classes according to the values of $Y$. In particular, let $S_1 = \{\boldsymbol{x}_i : Y_i < c\}$ and $S_2 = \{\boldsymbol{x}_i : Y_i \geq c\}$ for some constant $c$. We then calculate the optimal transport plan between $S_1$ and $S_2$, and repeat the process to obtain several plans. The displacement vectors based on these optimal transport plans are then pooled together to form the basis of the SDR subspace using the principal component analysis.

A number of questions remain unanswered, such as (1) how does the error in the response affect the result; (2) how does use other distance metrics in optimal transport instead of the $L_2$ norm affect the result; (3) would the multimarginal optimal transport approach [52] be a more appealing way to generalize the proposed method from binary-response to multi-class response than the one-vs-one strategy; (4) when the optimal couplings are obtained from the Sinkhorn algorithm, how does the Sinkhorn regularization parameter impacts the final dimension reduction; (5) as mentioned in [12], SIR with discretized response is closely related to linear discriminant analysis, then is there any relationship between the proposed method with the Wasserstein Discriminant Analysis approach [27, 43]. Additional research is needed to answer these questions and to better understand the proposed method.

Like many existing dimension reduction studies, POTD, by its nature, is a new methodology that aims to solve challenging high-dimensional problems. Hence, this work does not present any foreseeable societal consequence by itself. However, POTD has the potential to be applied to many high-dimensional data, e.g., imaging data, gene expression data, and so on. This work may speed up these researches and hence amplify the positive and negative impacts that exist in these scientific research fields.

## Acknowledgment

We highly appreciate the valuable comments and suggestions from anonymous reviewers. We thank the members of Big Data Analytics Lab, University of Georgia, for providing fruitful discussions. We would also like to thank Dr. David Gu for his insightful blog about the optimal transportation theory. This work was partially supported by National Science Foundation grants DMS-1903226, DMS-1925066, NIH grants R01GM113242, R01GM122080, NSFC grant 12001042, and Beijing Institute of Technology Research Fund Program for Young Scholars.

## Footnotes

[1]All the datasets are downloaded from UCI machine learning repository [6]

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
