[Supplementary Material]

# Supplementary Material to "Sufficient dimension reduction for classification using principal optimal transport direction"

Cheng Meng[1]  Jun Yu[2]  Jingyi Zhang[3]  Ping Ma[4]  Wenxuan Zhong[*4]

[1] Center for Applied Statistics, Institute of Statistics and Big Data, Renmin University of China
[2] School of Mathematics and Statistics, Beijing Institute of Technology
[3] Center for Statistical Science, Tsinghua University
[4] Department of Statistics, University of Georgia
chengmeng@ruc.edu.cn, yujunbeta@bit.edu.cn, jingyizhang@tsinghua.edu.cn, pingma@uga.edu,
wenxuan@uga.edu

## A  Proof of Theorem 1

*Proof.* Recall the definition of the second-order displacement matrix,

$$\boldsymbol{\Sigma} = \int \Big( (I - \phi^*)(X) \Big) \Big( (I - \phi^*)(X) \Big)^T d\mu(X).$$

The square of Wasserstein distance (with $L_2$ norm) between $X|Y^* = 0$ and $X|Y^* = 1$ can be written as

$$\int \mathrm{Tr} \Big\{ \Big( (I - \phi^*)(X) \Big) \Big( (I - \phi^*)(X) \Big)^T \Big\} d\mu(X) = \mathrm{Tr}(\boldsymbol{\Sigma}), \tag{1}$$

where $\mathrm{Tr}(\cdot)$ is the trace operator. In this proof, we restrict our attention on the optimal transport with $L_2$ norm. Therefore, the trace $\mathrm{Tr}\{((I - \phi)(X))((I - \phi)(X))^T\}d\mu(X)$, as a function of $\phi$, achieves its minimum value when $\phi = \phi^*$. Furthermore, for any $\boldsymbol{b} \in \mathbb{R}$, the term

$$\int \mathrm{Tr} \Big\{ \Big( \boldsymbol{b}^T (I - \phi^*)(X) \Big) \Big( \boldsymbol{b}^T (I - \phi^*)(X) \Big)^T \Big\} d\mu(X) = \boldsymbol{b}^T \boldsymbol{\Sigma} \boldsymbol{b},$$

can be regarded as the transportation cost via $\phi^*$ in the direction of $\boldsymbol{b}$.

Recall that $P(X) = E(Y|X)$, we have $Y^* = I(E(Y|X) \geq 0.5)$, which indicates $\mathcal{S}_{Y^*|X} \subseteq \mathcal{S}_{Y|X}$. Therefore, to prove Theorem 1, we only need to verify $\mathcal{S}_{Y^*|X} = \mathcal{S}(\mathbf{V}_r)$. Let $\mathbf{B}$ to denote a matrix that satisfy $Y^* \perp\!\!\!\perp X | \mathbf{B}^T X$. Without loss of generality, we assume $\mathcal{S}(\mathbf{B}) = \mathcal{S}_{Y^*|X}$. Let $\mathcal{S}(\mathbf{B})^\perp$ be the orthogonal complement space of $\mathcal{S}(\mathbf{B})$.

Recall that $\mathbf{V}_q = (v_1, \ldots, v_q) \in \mathbb{R}^{p \times q}$ is a matrix which contains orthonormal columns satisfying $\boldsymbol{\Sigma} v_j = \lambda_j v_j$ for $j = 1, \ldots, q$ with $\lambda_q > 0 = \lambda_{q+1}$. Therefore, the space spanned by $\boldsymbol{\Sigma}$ is equivalent to the space spanned by $\mathbf{V}_q$. Also note that $\mathrm{rank}(\mathbf{B}) = r$. If $\mathcal{S}(\mathbf{B}) = \mathcal{S}(\boldsymbol{\Sigma})$ holds, we have $\mathrm{rank}(\mathbf{B}) = \mathrm{rank}(\boldsymbol{\Sigma}) = r = q$, resulting in $\mathcal{S}(\mathbf{V}_r) = \mathcal{S}(\mathbf{V}_q) = \mathcal{S}(\boldsymbol{\Sigma}) = \mathcal{S}(\mathbf{B})$. The conclusion follows immediately. Hence, to prove Theorem 1, it is sufficient to show that $\mathcal{S}(\mathbf{B}) = \mathcal{S}(\boldsymbol{\Sigma})$ holds.

To verify $\mathcal{S}(\mathbf{B}) = \mathcal{S}(\boldsymbol{\Sigma})$, we only need to show the following two results hold:

(I). $\mathcal{S}(\boldsymbol{\Sigma}) \subseteq \mathcal{S}(\mathbf{B})$. That is to say, for any $\boldsymbol{b} \in \mathcal{S}(\mathbf{B})^\perp$, we have $\boldsymbol{b}^T \Sigma \boldsymbol{b} = 0$.

(II). $\mathcal{S}(\mathbf{B}) \subseteq \mathcal{S}(\boldsymbol{\Sigma})$. That is to say, for any $\boldsymbol{b} \neq \mathbf{0}$, such that $\boldsymbol{b} \in \mathcal{S}(\mathbf{B})$, we have $\boldsymbol{b}^T \Sigma \boldsymbol{b} > 0$.

We now begin with the statement (I). The high-level idea is that if there exists a $\boldsymbol{b} \in \mathcal{S}(\mathbf{B})^{\perp}$ such that $\boldsymbol{b}^T \Sigma \boldsymbol{b} > 0$, we can construct a new measure-preserving map with a smaller transport cost than $\phi^*$, resulting in a contradiction.

Without loss of generality, we assume $\mathbf{B} = (B_1, \ldots, B_r)$, $\mathbf{B}^{\perp} = (B_{r+1}, \ldots, B_p)$ with $B_i^T B_i = 1$ for $i = 1, \ldots, p$ and $B_i^T B_j = 0$ for $i \neq j$ and $i, j = 1, \ldots, p$. For any $\boldsymbol{b} \in \mathcal{S}(\mathbf{B})^{\perp}$, according to the definition of $\mathbf{B}$, we have $\boldsymbol{b}^T X \perp\!\!\!\perp Y^*$, i.e., $p(\boldsymbol{b}^T X | Y^* = 0) = p(\boldsymbol{b}^T X | Y^* = 1) = p(\boldsymbol{b}^T X)$. Moreover, we have

$$(\mathbf{B}^{\perp})^T X \perp\!\!\!\perp Y^*. \tag{2}$$

Let $\tilde{\phi}^{(1)} : \mathbb{R}^r \to \mathbb{R}^r$ be the optimal transport map from $\mathbf{B}^T X | Y^* = 1$ to $\mathbf{B}^T X | Y^* = 0$. We now construct a map from $(\mathbf{B}^{\perp}; \mathbf{B})^T X | Y^* = 1$ to $(\mathbf{B}^{\perp}; \mathbf{B})^T X | Y^* = 0$, denoted as $\tilde{\phi}^{(2)}(X) = (I_{p-r}; \tilde{\phi}^{(1)})(X)$. Combining (2) and the fact that $\tilde{\phi}^{(1)}$ is an optimal transport map, it is easy to check that $\tilde{\phi}^{(2)}$ is a measure-preserving map. Therefore, $\tilde{\phi}^{(2)} \circ (\mathbf{B}^{\perp}; \mathbf{B})^T (X)$ is a measure-preserving map from $X | Y^* = 1$ to $(\mathbf{B}^{\perp}; \mathbf{B})^T X | Y^* = 0$. Note that $(\mathbf{B}^{\perp}; \mathbf{B})(\mathbf{B}^{\perp}; \mathbf{B})^T = I_p$. Consequently, the map $\tilde{\phi}^{(3)}(X) = (\mathbf{B}^{\perp}; \mathbf{B})\tilde{\phi}^{(2)} \circ (\mathbf{B}^{\perp}; \mathbf{B})^T(X)$ is thus a measure-preserving map from $X | Y^* = 1$ to $X | Y^* = 0$. Let $\tilde{\Sigma}$ be second-order displacement matrix respecting to $\tilde{\phi}^{(3)}$. Recall that the first $r$ dimension in $\tilde{\phi}^{(2)}$ is an identity map between $\mathbf{B}^{\perp T} X | Y^* = 1$ to $\mathbf{B}^{\perp T} X | Y^* = 0$. We thus have

$$\begin{aligned}
&\mathrm{Tr}\left\{ (\mathbf{B}^{\perp})^T \tilde{\Sigma} \mathbf{B}^{\perp} \right\} \\
&= \int \mathrm{Tr}\left\{ \left(\mathbf{B}^{\perp T} X - I_{p-r}(\mathbf{B}^{\perp T} X)\right) \left(\mathbf{B}^{\perp T} X - I_{p-r}(\mathbf{B}^{\perp T} X)\right)^T \right\} d\mu(X) = 0. 
\end{aligned} \tag{3}$$

If there exists $\boldsymbol{b} \in \mathcal{S}(\mathbf{B})^{\perp}$ such that $\boldsymbol{b}^T \Sigma \boldsymbol{b} > 0$, we have

$$\mathrm{Tr}\left\{ (\mathbf{B}^{\perp})^T \Sigma \mathbf{B}^{\perp} \right\} > 0 \tag{4}$$

Note that $\tilde{\phi}^{(1)}$ is the optimal transport map from $\mathbf{B}^T X | Y^* = 0$ to $\mathbf{B}^T X | Y^* = 1$. This is to say, we have

$$\begin{aligned}
&\mathrm{Tr}\{\mathbf{B}^T \tilde{\Sigma} \mathbf{B}\} \\
&= \int \mathrm{Tr}\left\{ \left(\mathbf{B}^T X - \tilde{\phi}^{(1)}(\mathbf{B}^T X)\right) \left(\mathbf{B}^T X - \tilde{\phi}^{(1)}(\mathbf{B}^T X)\right)^T \right\} d\mu(X) \leq \mathrm{Tr}\{\mathbf{B}^T \Sigma \mathbf{B}\}. 
\end{aligned} \tag{5}$$

Combining the results in Equation (3), Inequality (4), and Inequality (5), we have

$$\begin{aligned}
&\mathrm{Tr}(\tilde{\Sigma}) \\
&= \int \mathrm{Tr}\left\{ \left((I - \tilde{\phi}^{(3)})(X)\right)\left((I - \tilde{\phi}^{(3)})(X)\right)^T \right\} d\mu(X) \\
&= \int \mathrm{Tr}\left\{ \left((\mathbf{B}^{\perp T}; \mathbf{B}^T)^T (I - \tilde{\phi}^{(3)})(X)\right)\left((\mathbf{B}^{\perp T}; \mathbf{B}^T)^T (I - \tilde{\phi}^{(3)})(X)\right)^T \right\} d\mu(X) \\
&= \mathrm{Tr}\left\{ (\mathbf{B}^{\perp})^T \tilde{\Sigma} \mathbf{B}^{\perp} \right\} + \mathrm{Tr}\{\mathbf{B}^T \tilde{\Sigma} \mathbf{B}\} \\
&< \mathrm{Tr}\left\{ (\mathbf{B}^{\perp})^T \Sigma \mathbf{B}^{\perp} \right\} + \mathrm{Tr}\{\mathbf{B}^T \Sigma \mathbf{B}\} \\
&= \int \mathrm{Tr}\left\{ \left((\mathbf{B}^{\perp T}; \mathbf{B}^T)^T (I - \phi^*)(X)\right)\left((\mathbf{B}^{\perp T}; \mathbf{B}^T)^T (I - \phi^*)(X)\right)^T \right\} d\mu(X) \\
&= \mathrm{Tr}(\Sigma).
\end{aligned} \tag{6}$$

Recall that the trace $\mathrm{Tr}\{((I-\phi)(X))((I-\phi)(X))^T\}d\mu(X)$, as a function of $\phi$, achieves its minimum value when $\phi = \phi^*$, which is the optimal transport map under the $L_2$ distance. Inequality (6) indicates $\tilde{\phi}^{(3)}$ is a measure-preserving map with a smaller transport cost than $\phi^*$, and thus makes contradiction. This completes the proof for Statement I.

It is worth mentioning that, for the optimal transport problem respecting other choices of distance metric, such as squared Euclidean distance, minimizing $\mathrm{Tr}(\Sigma)$ respecting $\phi$ does not necessarily lead

to the solution of $\phi^*$. Consequently, Inequality (6) may not leads to a contradiction in such cases. Therefore, our proof restricts our attention to the optimal transport with $L_2$ norm.

We then turn to Statement II. Suppose there is a $\boldsymbol{b} \neq \boldsymbol{0}$ lies in the space $\mathcal{S}(\mathbf{B})$ such that $\boldsymbol{b}^T \Sigma \boldsymbol{b} = 0$. Hence, there are exist at one $B_i$ for $i = 1, \ldots, r$ such that $B_i{}^T \Sigma B_i = 0$. Therefore,

$$\int \mathrm{Tr}\left\{\left(B_i^T(I - \phi^*)(X)\right)\left(B_i^T(I - \phi^*)(X)\right)^T\right\} d\mu(X) = 0. \tag{7}$$

Note that $\phi^*$ is a measure-preserving map. We have $B_i^T \phi^*(X)$ and $B_i^T X | Y^* = 0$ are identical distributed. From (7), it is clear to see that $B_i^T X | Y^* = 1$ and $B_i^T X | Y^* = 0$ are identical distributed. Hence, it follows that $p(B_i^T X | Y^* = 0) = p(B_i^T X | Y^* = 1) = p(B_i^T X)$, which implies that $B_i^T X \perp\!\!\!\perp Y^*$. Therefore, we have $\mathcal{S}(B_i, B_{r+1}, \ldots, B_p) \subseteq S_{Y^*|X}^{\perp}$. In other words, we have $S_{Y^*|X} \subseteq \mathcal{S}(B_1, \ldots, B_{i-1}, B_{i+1} \ldots, B_r)$. Hence we can conlude that the dimension of the central dimension reduction subspace $S_{Y^*|X}$ is less than $r - 1$. This leads to a contradiction with (H.2) where the structure dimension is $r$. $\qquad \square$

## B  Technical Lemmas

Before we prove Theorem 2, we introduce the following two technical lemmas, whose proof can be found in Theorem 4.1 in [1] and Section 3.3 in [2], respectively.

**Lemma 1.** *Assume $\mu, \nu$ are probabilities on $\mathbb{R}^p$ that both $\mu$ and $\nu$ have positive densities in the interior of their convex supports and have finite moments of order $4 + \delta$ for some $\delta > 0$. Let $X_1, \ldots, X_n$ are i.i.d., random variables with law $\mu$, $Y_1, \ldots, Y_m$ are i.i.d., random variables with law $\nu$, independent of the $X_i$'s, $\mu_n$ denotes the empirical measure on $X_1, \ldots, X_{n_0}$, and $\nu_m$ denotes the empirical measure on $Y_1, \ldots, Y_m$. As $n, m \to \infty$ and $n/(m + n) \to C \in (0, 1)$, it holds that*

$$\sqrt{n}(W_2^2(\mu_n, \nu) - EW_2^2(\mu_n, \nu)) \to N(0, \sigma^2(\mu, \nu)), \tag{8}$$

$$\sqrt{\frac{nm}{n+m}}(W_2^2(\mu_n, \nu_m) - EW_2^2(\mu_n, \nu_m)) \to N(0, (1 - C)\sigma^2(\mu, \nu) + C\sigma^2(\nu, \mu)), \tag{9}$$

*where $\sigma^2(P, Q) = \int(\|x\|^2 - 2\phi^*(x))^2 d\mu - (\int(\|x\|^2 - 2\phi^*(x))d\mu)^2$ and $\phi^*$ denotes an optimal transportation potential from $\mu$ to $\nu$.*

**Lemma 2.** *Let $\mu_n$ and $\nu_n$ be the empirical measure. Under (H.4), $EW_2^2(\mu_n, \mu) \leq Cn^{-2/p}$ and $EW_2^2(\nu_n, \nu) \leq Cn^{-2/p}$, where $C$ is some constant.*

## C  Proof of Theorem 2

*Proof.* For easy of presentation, let $\boldsymbol{x}_i$ be the $i$th observation of $X | P(X) \geq 0.5$, and $\phi^*, \hat{\phi}$ be the optimal transport map and the corresponding estimator, respectively. Let $x_i^{(j)}, X^{(j)}, \phi^{*(j)}(x_i), \hat{\phi}^{*(j)}(x_i), \phi^{*(j)}(X)$, and $\phi^{*(j)}(X)$ be the $j$th dimension of $x_i, X, \phi^*(x_i), \hat{\phi}^*(x_i), \phi^*(X)$, and $\phi^*(X)$, respectively, Note that under (H.3)–(H.5), the $(i, j)$-th entry of $\hat{\Sigma} - \Sigma$ satisfies

$$\|\hat{\Sigma}_{(i,j)} - \Sigma_{(i,j)}\|^2$$

$$= \left\{\frac{1}{n_1}\sum_{k=1}^{n_1}(x_k^{(i)} - \hat{\phi}^{(i)}(x_k))(x_k^{(j)} - \hat{\phi}^{(j)}(x_k)) - E[(X^{(i)} - \phi^{*(i)}(X))(X^{(j)} - \phi^{*(j)}(X))]\right\}^2$$

$$\leq 4\left\{\frac{1}{n_1}\sum_{k=1}^{n_1}x_k^{(i)}x_k^{(j)} - EX^{(i)}X^{(j)}\right\}^2 + 4\left\{\frac{1}{n_1}\sum_{k=1}^{n_1}\hat{\phi}^{(i)}(x_k)x_k^{(j)} - E\phi^{*(i)}(X)X^{(j)}\right\}^2$$

$$+ 4\left\{\frac{1}{n_1}\sum_{k=1}^{n_1}\hat{\phi}^{(j)}(x_k)x_k^{(i)} - E\phi^{*(j)}(X)X^{(i)}\right\}^2 + 4\left\{\frac{1}{n_1}\sum_{k=1}^{n_1}\hat{\phi}^{(i)}(x_k)\hat{\phi}^{(j)}(x_k) - E\phi^{*(i)}(X)\phi^{*(j)}(X)\right\}^2.$$

$$\tag{10}$$

Under (H.3), the first term in (10) is $O_p(n_1^{-1})$ due to the law of large number. Under (H.3)–(H.5), simple calculate yields,

$$\frac{1}{n_1} \sum_{k=1}^{n_1} \hat{\phi}^{(i)}(x_k) x_k^{(j)} - E\phi^{*(i)}(X)X^{(j)}$$

$$= \frac{1}{n_1} \sum_{k=1}^{n_1} \left( \hat{\phi}^{(i)}(x_k) - \phi^{*(i)}(x_k) + \phi^{*(i)}(x_k) \right) x_k^{(j)} - E\phi^{*(i)}(X)X^{(j)}$$

$$\leq \{\frac{1}{n_1} \sum_{k=1}^{n_1} (\hat{\phi}^{(i)}(x_k) - \phi^{*(i)}(x_k))^2\}^{1/2} \{\frac{1}{n_1} \sum_{k=1}^{n_1} (x_k^{(j)})^2\}^{1/2} + \frac{1}{n_1} \sum_{k=1}^{n_1} \phi^{*(i)}(x_k) x_k^{(j)} - E\phi^{*(i)}(X)X^{(j)}.$$

Note that the transportation cost between the empirical measure $\hat{\phi}(x_1), \ldots, \hat{\phi}(x_{n_1})$ and a given probability of the random variable $X|Y^* = 0$ equals to $EW_2^2(\nu_{n_0}, \nu) + O_p(n^{-1/2})$ according to Lemma 1. From Lemma 2, it is clear to see that $EW_2^2(\nu_{n_0}, \nu) = O_p(n^{-2/p})$ which implies that $\frac{1}{n_1} \sum_{k=1}^{n_1} (\hat{\phi}^{(i)}(x_k) - \phi^{*(i)}(x_k))^2 = O_p(n^{-2/p})$. Recall that $n_0/n_1$ goes to some constant bounded away from zero under (H.5). Therefore,

$$\frac{1}{n_1} \sum_{k=1}^{n_1} \hat{\phi}^{(i)}(x_k) x_k^{(j)} - E\phi^{*(i)}(X)X^{(j)}$$

$$\leq \{\frac{1}{n_1} \sum_{k=1}^{n_1} (\hat{\phi}^{(i)}(x_k) - \phi^{*(i)}(x_k))^2\}^{1/2} \{\frac{1}{n_1} \sum_{k=1}^{n_1} (x_k^{(j)})^2\}^{1/2} + \frac{1}{n_1} \sum_{k=1}^{n_1} \phi^{*(i)}(x_k) x_k^{(j)} - E\phi^{*(i)}(X)X^{(j)}$$

$$= O_p(n^{-1/p}) O_p(1) + O_p(n^{-1/2})$$

$$= O_p(n^{-1/p}).$$

Similarly, it can be shown that the last two terms in (10) are in the order $O_p(n^{-1/p})$. That is, $\|\hat{\Sigma} - \Sigma\|_F = O_p(n^{-1/p})$. Therefore, by the Davis-Kahan theorem [3] under (H.2), we have $\|\sin(V_r, \hat{V}_r)\|_F = O_p(n^{-1/p})$. □