[Reviews · NeurIPS 2020]

Review 1

Summary and Contributions: The paper introduces a new method for sufficient dimension reduction for discrete labels. It computes the class-wise optimal transport maps on Dirac particles concatenate the first-order displacement matrices of the maps. It then uses the principle direction of the concatenated matrix as the target to project the data onto. The paper provides the theoretical guarantee of lossless recovery of the data given mild conditions. The results from several experiments on synthetic and real data shows the method outperforms many other baseline methods.

Strengths: -- soundness of the claims (theoretical grounding, empirical evaluation) The theoretical derivation seems solid. -- significance and novelty of the contribution The idea of using the displacement matrices of optimal transport maps for dimensionality reduction seems novel. The theoretical hardness is yet low. -- relevance to the NeurIPS community This paper address linear dimension reduction which is a major and still open problem in the machine learning community.

Weaknesses: -- soundness of the claims (theoretical grounding, empirical evaluation) In my opinion, the evaluation is not thorough enough for the authors to claim that "the proposed POTD outperforms most of the state-of-the-art linear dimension reduction methods". The setup seems very easy. The improvement over other baselines in experiment 3 is marginal and yet the computational cost compared to the baselines is not available. -- 4 Algorithm It is not clear to me how the algorithm is related to the theories in 3. -- Line 248 I wouldn't call a simulation with n=400 and p=10, 20, 30 an "extensive" one. --Line 69 I can only find the motivation-ish of the work at Line 69-71, "... 'subspace robust' optimal transport plan [41]. We take one step forward and reveal the relationship between principle optimal transport directions and the SDR subspace." It is not clear that (1) why such a relation matters and motivated this work (2) if and how this work leverages the theoretical results from [41] -- I feel that the overall quality of the paper as presented in the current form is below the threshold but I am not an expert on sufficient dimension reduction and I am open to rebuttal.

Correctness: I haven't found any errors in terms of methodology.

Clarity: In terms of the language, yes, but it is not clear in several places that I pointed out in the rest of the comments.

Relation to Prior Work: I do not have enough knowledge to answer this question.

Reproducibility: Yes

Additional Feedback: Some questions for the purpose of the review. -- Is the paper connected to Wasserstein dictionary learning in some way? Schmitz, Morgan A., Matthieu Heitz, Nicolas Bonneel, Fred Ngole, David Coeurjolly, Marco Cuturi, Gabriel Peyré, and Jean-Luc Starck. "Wasserstein dictionary learning: Optimal transport-based unsupervised nonlinear dictionary learning." SIAM Journal on Imaging Sciences 11, no. 1 (2018): 643-678. -- Line 22 Is $q$ in (1) pre-defined and fixed? Do $q$ in Line 22 and $r$ in Line 185 denote the same thing? If $q$ or $r$, if they are interchangeable, equals 1, is it related to Max-sliced OT by Deshpande et al.? Deshpande, Ishan, Yuan-Ting Hu, Ruoyu Sun, Ayis Pyrros, Nasir Siddiqui, Sanmi Koyejo, Zhizhen Zhao, David Forsyth, and Alexander G. Schwing. "Max-sliced wasserstein distance and its use for gans." In Proceedings of the IEEE conference on computer vision and pattern recognition, pp. 10648-10656. 2019. -- Line 228 What are the time complexities of the other major baseline methods? -- Line 225 Does the regularization of the Sinkhorn algorithm has an impact on the principle direction of the OT map? -- Line 124 Can the authors provide a reference for the proof of the equivalence? Under what conditions does the equivalence hold? Do these conditions automatically hold for this paper? -- Line 269 "We shrink each 28 x 28 image to 14 x 14 using max-pooling and then stack it into a 196-dimensional vector." Why downsampling? Is the original dimensionality prohibits the evaluation? -- 204 "first-order displacement matrix" This is a naive question: does the discussion about the "second-order" in 3 guarantees the results about this "first-order" in 4? ----- After rebuttal: I have read the rebuttal and other reviewers' comments. Unfortunately, the authors dodged most of my questions that may hurt the paper and my concerns still stand. I suggest the authors improve the overall quality of their work. The motivation and several points raised by other reviewers needs improvement before the final evaluation and discussion of the results.


Review 2

Summary and Contributions: This paper presents a new methodology for sufficient dimension reduction with a categorical response variable, using optimal transport theory. The key idea of the paper is to compute the empirical optimal transport coupling between samples belonging to different categories. The top eigenvectors of the optimal transport displacement matrices form a natural proxy for the sufficient dimension reduction subspace, and this fact is formally proved by the authors in Theorem 1, under certain conditions. Beyond this theoretical contribution, the authors discuss numerical considerations, and perform a thorough simulation study and real data analysis, showing the favourable performance of their method.

Strengths: The use of optimal transport for assessing the directions which carry most information for distinguishing between categories is a very natural idea, and I am glad the authors have noticed it. Their method is novel and simple to describe. Advances in computational optimal transport also make this method relatively tractable to compute. The problem addressed by the authors is well-motivated from the outset, and I appreciate their illustrative example in the introduction. Their approach is also proved to achieve sufficient dimension reduction in some settings, by Theorem 1. Overall, the paper is very readable, and provides a new approach to a problem which is certainly of interest to the NeurIPS community. This paper also raises interesting directions which could be explored in future work. Here are three examples which came to my mind. (i) The algorithm itself can trivially be extended to optimal transport costs which are non-Euclidean. Clearly this would greatly alter the resulting dimension reduction, and it would be of interest to understand its performance. It would also be of interest to understand how the result of Theorem 1 would change. (ii) When there are k=2 classes, the algorithm is naturally built upon the classical optimal transport problem between the two distributions. When k > 2, the authors suggest extending the method by computing all pairwise optimal transport problems between the k samples. It seems to me that a more natural approach would be to use the multimarginal optimal transport problem, which is a generalization of the optimal transport problem between more than two measures. (iii) When the optimal couplings used numerically are obtained from the Sinkhorn algorithm, it would be of interest to understand the extent to which the Sinkhorn regularization parameter impacts the final dimension reduction. There are likely many other points of discussion worth making about this method, and the authors might consider adding a Discussion section to their paper. If they choose to do so but are out of space, my personal opinion is that the toy example in lines 208-223 can be removed to make space.

Weaknesses: The primary weaknesses of this paper are outlined in the following points, for which I would appreciate the authors' response. 1. The optimal transport map estimator \hat \phi^* on line 183 is not formally defined, if I am not mistaken. In the case where the sample sizes of both classes are equal, there is a natural transport map estimator since the Monge problem is feasible. However, there does not exist a transport map between two empirical measures with different sample sizes, thus a transport map estimator cannot simply be obtained by the plug-in principle. Please clarify these aspects before stating Theorem 2. 2. In the proof of Theorem 2, the rate O_p(n^{-1/p}) for the optimal transport map error in the third-to-last-line of the display before line 57 is nontrivial to obtain, to my knowledge. I am not aware of a result yielding this rate under the conditions stated in the Theorem. This rate can be obtained for Wasserstein distances between the empirical measure and the truth, as in the references given, but this does not trivially translate into a rate for the quantity the authors bound here. If I am mistaken about this, I would appreciate a more precise reference from the authors, or a proof. 3. The proof of Theorem 2 also contains some minor technical mistakes and/or typos. I believe the rate O_p(n_1^{-2}) on line 55, and the rate O_p(n_1^{-1}) in the second-to-last line of the display before line 57, should be O_p(n_1^{-1}) and O_p(n_1^{-1/2}) respectively, by the Central Limit Theorem. Furthermore, in equation (6), one of the \hat \phi^{(j)} should be replaced by \hat \phi^{(i)}, I believe. I would not be opposed to the authors removing Theorem 2 entirely. Theorem 1 contains the key result about the method. 4. The main method described in the paper, lines 152-158, is hard to find. I personally would have appreciated seeing this earlier on in the paper, possibly isolated in its own subsection, and not necessarily combined with the technical aspects at the top of page 4 which are not all needed for describing the method. ---------- Comments following rebuttal. I would like to reiterate my interest in the methodology proposed by the authors, and I would be happy to see it published. The authors' rebuttal has addressed points (1), (3), (4) satisfactorily. My concern remains with point (2), since the authors have not provided specific references in their rebuttal. I would point the authors to [arXiv:1905.05828] and references therein for L^2 guarantees for certain transport map estimators, and encourage them to either update the Theorem with appropriate references, or to simply state as an assumption that the transport map estimator is consistent with the suitable rate. While it would be ideal to verify these modifications in a second round of revision, I maintain that they are minor in scope compared to the main contributions of the paper, and thus do not alter my score, under the assumption that the authors will make the appropriate modifications.

Correctness: As mentioned above, the proof of Theorem 2 contains mistakes which need to be addressed. I have not found mistakes in the proof of Theorem 1. The empirical methodology used in the paper seems broadly correct. However, one key point which I was unable to find was the choice of algorithm the authors use for computing the empirical optimal couplings for their method. On lines 224-229, they mention several potential algorithms that could be used. If they used Sinkhorn, what was their choice of regularization parameter? These points make their study non-reproducible for now.

Clarity: The paper is well-written, though as I have suggested above, it might improve clarity if Section 3 were restructured. Minor grammatical and stylistic suggestions are included below.

Relation to Prior Work: Yes, the distinction between this work and past literature is made clear. The background on sufficient dimension reduction and optimal transport is sufficiently thorough, in my opinion, and the authors' approach is compared against a large number of competing methods in the simulation study.

Reproducibility: Yes

Additional Feedback: The notation P(X) for E(Y|X) is confusing, I encourage the authors to use a different shorthand. The following are very minor grammatical and/or stylistic suggestions. - On line 33, replace "conditions that imposed" with "conditions that are imposed". - On line 34, remove "illustrative" as it is repetitive. - On line 105, remove the quotes on "distance". - On lines 114-115, "it may fail to find a solution" is confusing. The authors may simply write that the set \Phi is empty in some cases, hence the Monge problem is infeasible. - On line 122, "Brenier's" needs no apostrophe. - On line 127, since the reader has not yet been told the methodology, it is confusing to write Y \in \{0,1\} at this stage. When I first read this, I thought the entire methodology of the paper would be for this case. - On line 190, there should be a comma after \hat \Sigma. - On lines 205-206, "the matrix which including all the \Delta_{ij}" is confusing. First, "including" should be "includes". Second, the definition of \Gamma is imprecise. Since its precise definition appears in the algorithm, I at least suggest writing the dimension of \Gamma in the text here. - On line 323, should "penitential" read "potential"? - Supplement line 37: "Combine" should be "Combining". - Supplement line 39: "Contradictory" should be "contradiction".


Review 3

Summary and Contributions: This paper deals with dimensionality reduction: it aims to find a linear, lower dimension subspace of the features that can capture the same amount of predictive power as the original feature space. The paper assumes that the response variable has finite cardinality. In this case, the authors propose to find the displacement matrix computed from the optimal transport between the conditional distributions (see eq. (6)). The proposed subspace is the eigensubspace of the highest eigenvalues. The authors show that their method performs competitively in the numerical experiments.

Strengths: 1. I personally find the research question interesting. 2. The numerical section show that the proposed method works well empirically.

Weaknesses: 1. I am still not clear about the contribution of this paper. The final goal of this paper is to estimate the subspace S_{Y*|X}, and many existing papers already compute this subspace S_{Y*|X}. The result in this paper only provides an alternative to compute this subspace using ideas from optimal transport. It is unclear to me how different methods to compute the same object can lead to better results on the downstream task after being properly adjusted for numerical errors. Unfortunately, throughout the whole paper, the authors do not provide any explanation/intuition on why their method can systematically perform better than the competitors. I thus feel skeptical about the scientific contribution of this paper, and I still wonder how POTD can outperform in Table 1. 2. There is a huge gap between Section 3 and Section 4. The main idea of Section 3 is for binary output, however, Section 4 generalizes to k-ary output. This generalization is not justified and not well explained: - what is the role of the matrix Lambda? - what are the properties of Lambda (dimensions, symmetry, etc.)? - why should the r singular vectors of Lambda will span S_{Y*|X}?

Correctness: I don't find any trivial error mathematical error in the paper, except for minor typos. The generalization to k-ary output in Section 4 may not be correct because the authors do not provide any formal proof.

Clarity: - Section 3 and 4 can be significantly improved, as mentioned in the weakness part. - In equation (6), Sigma is computed using the optimal transport from mu to nu. What happens if we compute Sigma using the optimal transport from nu to mu? Will we recover the same result for any fixed value of r? - I don't fully understand what the authors want to convey with Figure 2, as well as the paragraph in lines 208-223. How can they justify that their method is more "effective" without any measure of performance? - In Algorithm 1, the assignment of Lambda_(i) has an index mismatch with j - I think the authors are confused between two words "principle" (used in the title) and "principal" (used in the rest of the paper).

Relation to Prior Work: The authors provide sufficient discussion on the literature.

Reproducibility: Yes

Additional Feedback: - In line 184, should the expression of Sigma be dependent on X_{(1)}, instead of X? - Line 259: should it be I_p instead of I_q? %% Comments added after the rebuttal I thank the authors for their rebuttal. Unfortunately, I am unwilling to increase my score for the following reasons: 1. There is no clear statement on the contribution of this paper that makes it stand out from the literature. The authors reply to my concern about the contributions in point 1:(1) using highly ambiguous terms which are not defined in the paper (what do "full SDR mean subspace" and "partial SDR mean subspace" mean?). To resolve this issue, I suggest the authors provide strong arguments to support the optimal transport approach. From the statistical viewpoint, possible arguments include (but do not limit to) theoretical advantages of the OT method (does it have faster convergence rate? does it have better finite-sample guarantee?, or something similar?). From computational viewpoint, possible arguments include a full comparison of algorithmic complexity. Clearly, the numerical section indicates that there are at least 15 other methods that perform SDR. The numerical result does not show that the OT approach outperforms the rest, as such, more theoretical justification is needed to support the OT approach. 2. The paper is not coherent: the theory is for 2-ary, the numerical result is for k-ary. The authors should either run the numerical results for 2-ary, or develop a complete analysis for k-ary classification. 3. The other referees have raised their concerns about certain technical details of the paper. Unfortunately, the authors do not provide a precise technical fix for these concerns. I fear that the paper has a considerable risk of containing false theoretical claims.


Review 4

Summary and Contributions: - the authors considered sufficient dimension reduction problem for classification - the authors proposed a method based on optimal transport to estimate a sufficient dimension reduction space, performed an extensive testing and compared the method with different baseline approaches

Strengths: - a new method for sufficient dimension reduction in case of categorical output - the empirical evaluation section contains a lot of experiments and clearly demonstrates efficiency of the proposed method compared to baseline solutions - the authors proposed a theoretical justification for the developed approach

Weaknesses: - the authors mentioned that there are two methods for linear sufficient dimension reduction, namely, [29,46]. Why the authors did not compare their results with these approaches? - the authors did not discuss limitations of the proposed approach. There are conditions H1-H5 under which the authors proved convergence of the algorithm. However, are these conditions necessary? In which situation the proposed approach does not work?

Correctness: - what if the considered classification problem is imbalanced? How does the imbalance ratio influence results of the sufficient dimension reduction? - in figure 3 the authors compared visualisation based on PCA with the visualisation used on the proposed methods. What is about similar illustrations for other DR methods? E.g. ISOMAP? Such figures can be provided in the appendix.

Clarity: - 187-188: some problems with bold font of \hat{V}_r in the displayed formula - how can be the proposed method generalised to the case of multi-class classification? - quality of many figures is very low. It is difficult to understand the contents of figure 4.

Relation to Prior Work: - I think that the relation to prior work is clearly discussed.

Reproducibility: Yes

Additional Feedback: I read carefully comments of other reviewers. I decided to decrease my grade for one point, although still I think that it is OK if the paper is accepted: the paper proposes new interesting approach based on OT and has some theoretical justification although it is only for 2-class case. At the same time I agree with reviewers, that 1) the authors need to check more carefully the proof the Theorem 2 (due to the comment and concern about the used conditions, which I missed initially), 2) computational costs should be discussed, etc.

[Author Response · NeurIPS 2020]

We would like to thank the meta reviewer and four reviewers for the care with which you handle the submission and for
your professional and constructive comments. We have made every effort to address the concerns.

**Response to Reviewer 1's comments**
1 **Weaknesses**: (1) To the best of our knowledge, our work is the first approach that can achieve a "full SDR mean
subspace" instead of a partial SDR mean subspace obtained by alternative approaches. Theorem 1 provides a
theoretical justification for the success of our method's empirical performance, as all the alternatives can only
recover part of the SDR mean subspace, which may be far away from the true one. (2) The classification problem
in Experiment 3 is a favorable case for dimension reduction; thus, it warrants the asymptotic convergence of all
dimension reduction methods. We use such a problem to demonstrate that even for such a simple setup, our
method can work as well as any other methods. Furthermore, as demonstrated in the simulation that does not favor
conventional dimension reduction approaches, our method significantly outperforms all the other approaches. (3) We
will add a comparison of CPU time in the revision. (4) We extended the proposed method for binary response cases
(illustrated in line 152-157), of which the consistency of our SDR estimates is ensured in Theorem 1, to multi-class
response cases, detailed in Algorithm 1. (5) The $\Lambda \in \mathbb{R}^{nk \times d}$ in Algorithm 1 is used to generate the displacement
matrix of OTP for multi-class response cases using one-vs-rest strategy. We found using the one-vs-one strategy
yields similar results. Theorem 1 ensures that the right singular vector of $\Lambda$ is consistent for the SDR subspace.
We will provide more discussion of theoretical properties of $\Lambda$. (6) Both [41] and our method aims to optimize
some objective functions with respect to the displacement matrix. However, [41] focuses on finding a surrogate of
displacement matrix for OT such that the estimate of Wasserstein distance is robust, while our method aims to find
the SDR subspace using OT. Thus, the theoretical results of our work and [41] rooted in completely different lands.
2 **Additional feedback**: (1) The Wasserstein dictionary learning is an unsupervised approach; however, our method is
a supervised one. They are not directly related. (2) The $q$ is the true dimensionality of SDR space, while $r$ is the
estimated dimensionality. If $Y \perp\!\!\!\perp X | Y^*$, we can show $q = r$, otherwise $q \geq r$. Thus, they are not exchangeable. (3)
Slicing in our method is not used to estimate OT but the SDR. Thus, these two approaches are not related. (4) The
time complexities will be added. (5) We agree that that the effect of the regulation parameter may play an essential
role in OT estimate. However, in our paper, we used the EMD method that does not include a regulation parameter.
We may include this type of study in our future publications. (6) We will cite the Brenier theorem that demonstrates
the equivalence of Monge and the Kantorovich formulations under certain conditions, which are not required for our
work. (7) Max-pooling is used only for numerical convenience. The accuracy will not change significantly without
down-sampling. (8) Yes. These two estimates are the same.

**Response to Reviewer 2's comments**
3 **Strengths**: We highly appreciate the insightful comments and future-research suggestions.
4 **Weaknesses**: (1) We apologize for the confusion. Our theorem only require $\hat{\phi}^*$ in line 183 to be a consistent estimator
for OT. The existence of the Monge map is not required. We have revised the manuscript accordingly. (2) We will
include more detailed discussion and corresponding references of the OT convergence rate in the revision. (3-4)
Suggestions will be taken.
5 **Correctness**: We used the EMD method in python `POT` package instead of Sinkhorn to calculate the OTP. Thus, the
results are reproducible. More details will be provided.
6 **Reproducibility**: We will upload codes with detailed comments to NeurIPS and GitHub.
7 **Additional feedback**: Typos will be corrected. Suggestions will be taken.

**Response to Reviewer 3's comments**
8 **Weaknesses**: (1) Please see 1:(1) and 1:(4)–(5) for your weakness concerns.
9 **Clarity and Additional feedback**: (1) All results are not susceptible to the order of the transport or label switching.
(2) Fig.2 illustrates how our proposed method outperformed the conventional moment-based SDR approaches, which
estimate the space that is sufficient for conditional moments of $X|Y$. For example, SIR estimates SDR using the first
conditional moments while SAVE estimates SDR using the second conditional moment. More discussion will be
included in the revised manuscript. (3) Typos have been corrected.

**Response to Reviewer 4's comments**
10 **Weaknesses**: (1) Suggestion will be taken. (2) (H.1)–(H.2) in Theorem 1 are necessary conditions for the consistency
of the estimated SDR subspace. Violations of the conditions will forfeit the consistency of any SDR methods. For
example, if the SDR subspace is not unique as required by (H.1)–(H.2), the true SDR subspace is not well defined,
not to mention the consistency of the estimated SDR subspace. However, Conditions (H.3)–(H.5) are only sufficient
conditions which ensure the convergence of the empirical OT.
11 **Correctness**: (1) When data are extremely imbalanced, (H.5) is violated because the number of observations
respecting to different classes is not at the same order, in which case Theorem 2 will fail. However, this problem can
be fixed by adding weight to each of the classes when calculating the SDR directions. Such a generalization is not
trivial and is beyond the scope of the current manuscript. (2) Suggestions will be taken.
12 **Clarity**: (1) The typo will be fixed. (2) Please see 1:(4)–(5) for your concern. (3) We apologize for the low quality of
the figures. High-resolution figures will be used in the revision.

[Meta-Review · NeurIPS 2020]

Two reviewers support accept, and two reviewers indicate reject. I've looked at the paper, reviews and rebuttal, and recommend an accept decision based on the significance, theoretical grounding and novelty of the contribution. It is refreshing to see a new approach to a problem relevant to the NeurIPS community. However, please revise the paper to provide better explanation of terminology and intuition as to why the method works; the reviewers' additional feedback and post-rebuttal comments should also be carefully considered in the revisions.